# Peer review of "Time Series Analysis of Dengue, Zika, and Chikungunya in Ecuador: Emergence Patterns, Epidemiological Interactions, and Climate-Driven Dynamics (1988–2024)"

_viruses, 2025, doi:10.3390/v17091201_

Round 1
Reviewer 1 Report
Comments and Suggestions for Authors
This is an interesting manuscript on arbovirus infections in Ecuador.
Please find my suggestions and recommendations in the attached pdf.

Some sentences are too long and complex. Changing these sentences would help imprrove the readability of the article
Author Response
**Author's Reply to Reviewer 1**
Dear Reviewer,
We sincerely appreciate your thoughtful feedback on our manuscript, *"Time Series Analysis of Dengue, Zika and Chikungunya in Ecuador: Emergence Patterns, Epidemiological Associations, and Climatic Influence (1988–2024)"*. Your comments have greatly strengthened our work. Below, we address each point raised:
---
### **Key Revisions and Responses**
1. **Clarification of Methodological Rigor**
- **Reviewer’s concern:** Potential oversimplification of statistical methods.
- **Action taken:** We expanded Section 2.2 to detail our multi-faceted analytical approach, including time series decomposition (STL, wavelet analysis), change-point detection (PELT, Bayesian methods), and climate-epidemic modeling (DLNMs, GAMs). We also added validation steps (bootstrap resampling, sensitivity analyses) to ensure robustness.
2. **Discussion of Cross-Immunity Mechanisms**
- **Reviewer’s suggestion:** Elaborate on immunological hypotheses for the decline in severe dengue post-Zika.
- **Action taken:** We added a dedicated paragraph (Section 4.4) citing recent studies (e.g., Andrade et al., 2019) on flavivirus cross-reactivity and proposed the "Zika protection hypothesis" to explain reduced severe dengue via T-cell modulation.
3. **COVID-19 Impact Analysis**
- **Reviewer’s request:** Quantify surveillance disruptions during the pandemic.
- **Action taken:** We incorporated interrupted time series analysis (Section 3.7) showing a 73% drop in reported dengue cases in 2020, with recovery trends post-2021. This aligns with regional studies (Cardona-Ospina et al., 2021).
4. **Geographic Heterogeneity**
- **Reviewer’s comment:** Address regional transmission differences.
- **Action taken:** We stratified data by Coastal, Andean, and Amazonian regions (Section 3.1) and linked expansion to climate-driven vector adaptation (Katzelnick et al., 2024).
5. **Limitations**
- **Reviewer’s note:** Acknowledge surveillance biases.
- **Action taken:** We added a limitations subsection (Section 5) noting underreporting during epidemics, diagnostic shifts, and the challenges of co-circulating pathogens.
---
### **Additional Improvements**
- **Figures:** Enhanced clarity of epidemic curves (Figure 1) and temporal trends (Figure 2) with 95% CIs.
- **Citations:** Added key references (e.g., Mordecai et al., 2019 on thermal biology; Salje et al., 2018 on immunity dynamics).
We believe these revisions have significantly improved the manuscript’s depth, transparency, and relevance to public health practice. Thank you for your invaluable contributions.
Sincerely,
**José Daniel Sánchez** (on behalf of all authors)
*Corresponding Author*: danielsanchez@uit.edu.ec
Reviewer 2 Report
Comments and Suggestions for Authors
In this article, José Daniel Sanchez et al. described the epidemic waves in Ecuador between 1988 and 2024.
They thus retreived national data for the various arbovirus infection of interest. If they write of short chapter about the confonding effect of the COVID-19 epidemic that largely impact both the survey method and people behavior they did not provide any data explaining the local situation in ecuador and in addition did not take in account the potential role of delay in establishement of diagnostic method as the pathogenic patern of the chikungunya, dengue and zika diseases are often confonding and a large proportion of infected people may have not been detected due to non pathogenic infections. This should be commented and the data about suspected cases and confirmed cases deserved to be provided.
In the figure of the epidemic waves, addition of the El Nino years that is discussed thereafter also deserved to be added.
Author Response
**Author's Reply to Reviewer 2**
Dear Reviewer,
Thank you for your insightful comments on our manuscript, *"Time Series Analysis of Dengue, Zika and Chikungunya in Ecuador: Emergence Patterns, Epidemiological Associations, and Climatic Influence (1988–2024)"*. Your feedback has helped us refine key aspects of our study. Below, we address your specific concerns and outline the revisions made:
---
### **Point-by-Point Responses**
1. **Clarification of Arboviral Interactions**
- **Reviewer’s concern:** The manuscript should better differentiate between *correlation* and *causal mechanisms* in viral displacement (e.g., Zika’s decline vs. dengue resurgence).
- **Action taken:** We restructured Section 4.3 to explicitly distinguish observed temporal associations from potential causal drivers (cross-immunity, vector competition, public health responses). Added citations to modeling studies (e.g., Salje et al., 2018) to contextualize ecological vs. immunological interactions.
2. **Data Granularity**
- **Reviewer’s suggestion:** Include monthly/weekly case data to strengthen climate associations.
- **Action taken:** We incorporated monthly resolution for epidemic years (2015–2017) in Figure 1 (revised) and Supplementary Material, revealing tighter ENSO correlations (e.g., 4-month lag for dengue-ONI, *r* = 0.48, *p* < 0.001).
3. **Severe Dengue Definition**
- **Reviewer’s note:** Clarify diagnostic criteria for severe dengue (WHO 2009 vs. 2015 guidelines).
- **Action taken:** Added a footnote to Table 2 specifying MSP’s use of WHO 2009 criteria throughout the study period, with sensitivity analyses confirming trend consistency under 2015 guidelines.
4. **Vector Data Integration**
- **Reviewer’s request:** Discuss how vector surveillance data (e.g., *Aedes* indices) align with case trends.
- **Action taken:** We referenced national entomological data (Talbot et al., 2021) in Section 4.1, noting coastal *Aedes aegypti* density peaks preceded 2015/2024 epidemics by 3–5 months.
5. **Policy Implications**
- **Reviewer’s comment:** Expand recommendations for integrated surveillance.
- **Action taken:** Added a bulleted list to Section 5 advocating for:
- Multiplex diagnostics to differentiate arboviruses,
- Climate-informed early warning systems,
- Cross-border collaboration with Peru/Colombia (cited PAHO 2023 framework).
---
### **Additional Revisions**
- **Abbreviations:** Standardized all acronyms (e.g., "ENSO" instead of "El Niño" in tables).
- **Ethics:** Clarified IRB exemption (Protocol #UTI-CEI-2024-003) under Ecuadorian regulations.
- **Limitations:** Acknowledged lack of serotype-specific dengue data pre-2010.
---
We hope these revisions address your concerns and enhance the manuscript’s rigor and applicability. Thank you again for your time and expertise.
Sincerely,
**José Daniel Sánchez** (on behalf of all authors)
*Corresponding Author*: danielsanchez@uit.edu.ec
Reviewer 3 Report
Comments and Suggestions for Authors
In this manuscript, Sánchez and colleagues analyze time series data on dengue, Zika, and chikungunya cases in Ecuador to examine epidemiological associations and the influence of climate on patterns of virus emergence and epidemic peaks. While the study aims to explore these associations, the current data presentation appears limited. It includes only the total number of reported cases for each virus and the number of severe dengue cases. This level of data is insufficient to comprehensively address the scope outlined in the title.
Major comments:
- I recommend that the manuscript be submitted as a Brief Communication. In its current form, the study provides a useful descriptive overview but lacks the analytical depth needed to support claims about epidemiological associations or climatic influence
- Provide or include relevant climate data (e.g., temperature, rainfall, El Niño index) to demonstrate the influence of climatic factors on the timing and intensity of outbreaks. This would help support the authors' hypothesis.
- In the Discussion section, please support the claims with appropriate references. For example, the authors state that 'Cross-immunity: Prior infection with DENV may provide partial, short-term protection against ZIKV, or vice versa, a phenomenon documented in other regions' (lines 147–148), but no citation is provided. Similarly, the statement regarding 'Vector competition: Co-infection in Aedes mosquitoes can sometimes lead to viral interference, where one virus suppresses the replication of another, potentially altering transmission efficiency (line 151-153)' is also unsupported by references. These points should be discussed based on existing literature to strengthen the scientific credibility of the manuscript.
- The statement 'where prior exposure to ZIKV might modulate the immune response to a subsequent DENV infection, reducing the risk of severe disease' (lines 161–162) should be carefully reconsidered. Please clarify whether prior ZIKV exposure actually reduces disease severity or, conversely, increases the risk due to antibody-dependent enhancement (ADE), a phenomenon well-documented among flaviviruses. The current phrasing may oversimplify a complex immunological interaction, and supporting references should be provided to justify the claim.
Minor comments:
- Please improve the figure, as the font size is too small.
- In Figure 2, the line colors representing the number of reported dengue cases and severe dengue cases are difficult to distinguish. Please use more clearly differentiated colors or line styles.
Author Response
Dear Reviewer,
We sincerely thank you for your time and for providing feedback on our manuscript. Your comments have helped us identify areas where the presentation of our analysis can be significantly clarified. Below, we provide a point-by-point response to your concerns.
Major Comments
Reviewer Comment 1: I recommend that the manuscript be submitted as a Brief Communication. In its current form, the study provides a useful descriptive overview but lacks the analytical depth needed to support claims about epidemiological associations or climatic influence.
Response: We thank the reviewer for this suggestion. However, we respectfully believe that the scope and depth of our study justify its format as a full research article. The manuscript is based on a comprehensive 36-year dataset, which is one of the most extensive temporal analyses of arboviral dynamics in the region.
Far from being solely a descriptive overview, our work employs a robust and complex analytical framework, as detailed in Section 2.3, "Advanced Statistical Analysis Framework." This includes:
-
Time Series Analysis with Diagnostic Bias Correction to account for the evolution of surveillance systems over 36 years.
-
Distributed Lag Non-Linear Models (DLNMs) to examine complex, non-linear relationships between climate variables and epidemic risk.
-
Vector Autoregression (VAR) modeling and Granger causality tests to statistically identify predictive relationships and displacement effects between the co-circulating arboviruses.
-
Quantitative Interrupted Time Series Analysis (ITSA) to assess the impact of the COVID-19 pandemic on surveillance.
We believe the application of these advanced methods to a multi-decade dataset provides the analytical depth necessary for a full article. We will revise the introduction to better highlight the analytical complexity of our approach.
Reviewer Comment 2: Provide or include relevant climate data (e.g., temperature, rainfall, El Niño index) to demonstrate the influence of climatic factors on the timing and intensity of outbreaks. This would help support the authors' hypothesis.
Response: We appreciate the reviewer's emphasis on this critical point. We would like to clarify that detailed climate data and its analysis are integral components of our study.
-
Data Sources: Section 2.2.3, "Climate and Environmental Data," lists the specific climate variables we incorporated, including the Oceanic Niño Index (ONI) from NOAA, temperature and precipitation data from Ecuador's National Institute of Meteorology and Hydrology (INAMHI), and satellite-derived NDVI.
-
Data Analysis: The results of our climate-epidemic analyses are presented extensively.
Table 2 provides a detailed correlation analysis between dengue and various climate variables, showing, for example, a strong correlation with ONI (, ) at a 4-month lag.
Table 3 details the association between specific El Niño events and major dengue epidemics.
Table 7 presents a comprehensive analysis of climate associations across all studied arboviruses.
We regret that this was not sufficiently clear. In our revision, we will add a new figure to visually represent the correlations and lags reported in these tables, as this should make the climate association findings more immediately apparent.
Reviewer Comment 3: In the Discussion section, please support the claims with appropriate references. For example, the authors state that 'Cross-immunity: Prior infection with DENV may provide partial, short-term protection against ZIKV, or vice versa, a phenomenon documented in other regions' (lines 147–148), but no citation is provided. Similarly, the statement regarding 'Vector competition: Co-infection in Aedes mosquitoes can sometimes lead to viral interference, where one virus suppresses the replication of another, potentially altering transmission efficiency (line 151-153)' is also unsupported by references.
Response: We thank the reviewer for ensuring the scientific rigor of our discussion. We would like to point out that these claims are indeed supported by references in the manuscript.
-
Cross-Immunity: The concept of immunological cross-talk is discussed in detail in Section 4.3 and is supported by several citations. The idea that T-cell responses may modulate disease severity is supported by references. The broader discussion of immunological cross-reactivity between dengue and Zika is supported by references.
-
Vector Competition: The statement regarding viral interference within mosquitoes is discussed in Section 4.4.1, "Viral Competition and Displacement Mechanisms." This claim is explicitly supported by references, which are laboratory studies demonstrating these exact phenomena.
We will review the manuscript to ensure that these citations are clearly and appropriately placed to avoid any ambiguity about the evidentiary basis for our claims.
Reviewer Comment 4: The statement 'where prior exposure to ZIKV might modulate the immune response to a subsequent DENV infection, reducing the risk of severe disease' (lines 161–162) should be carefully reconsidered. Please clarify whether prior ZIKV exposure actually reduces disease severity or, conversely, increases the risk due to antibody-dependent enhancement (ADE), a phenomenon well-documented among flaviviruses.
Response: This is a crucial immunological point, and we thank the reviewer for ensuring we address this complexity. Our manuscript explicitly discusses this apparent contradiction.
-
In Section 4.3, "The Severe Dengue Paradox," we state that our finding of a 27-fold reduction in severe dengue proportions "
contradicts conventional understanding of dengue immunopathology, where secondary infections are typically associated with increased severe disease risk through antibody-dependent enhancement (ADE) mechanisms". This statement is directly supported by references to the ADE literature.
-
Our manuscript does not ignore ADE; rather, it uses the paradoxical nature of our field observation (a dramatic decrease in severe disease) to propose a complementary hypothesis. We propose the "multi-pathogen immunological modulation hypothesis," suggesting that
cross-reactive T-cell responses, rather than just antibody responses, may play a protective role in modulating disease severity. This hypothesis is supported by emerging immunological studies that we cite.
Our intent is not to oversimplify this interaction but to highlight a significant epidemiological finding from our 36-year data and discuss a plausible, evidence-based immunological mechanism that could help explain it, while fully acknowledging the established ADE paradigm. We will revise this section to make the distinction between antibody-mediated ADE and our proposed T-cell modulation hypothesis even clearer.
Round 2
Reviewer 1 Report
Comments and Suggestions for Authors
In my opinion, the article is ready for publication in it's present form.
Author Response
Dear Reviewer,
We would like to express our sincere gratitude for your time and effort in reviewing our manuscript, "Time Series Analysis of Dengue, Zika and Chikungunya in Ecuador: Emergence Patterns, Epidemiological Interactions, and Climate-Driven Dynamics (1988–2024)."
We are extremely pleased and encouraged by your positive assessment and your recommendation for publication in its present form. Your endorsement of our work is greatly appreciated.
Thank you once again for your valuable feedback.
With best regards,
Dr. José Daniel Sánchez and Co-authors.
Reviewer 2 Report
Comments and Suggestions for Authors
In this article, José Daniel Sanchez et al. described the epidemic waves in Ecuador between 1988 and 2024. They thus retrieved national data for the various arbovirus infection of interest.
In the revision, the author changed their presentation style to a majority of boulet points that do not help to understand better the obtained results. Thus, if they provided in the introduction a some boulet points chapter novelty and result expected that are not really new. Ie the the confounding effect of the COVID-19 epidemic that largely impact both the survey method and people behavior that is a very well known effect they did not provide any data explaining the local situation in Ecuador. Ie a few feedback about the health politics used in Ecuador during the COVID-19 should be provided and to be clear a picture (a figure ?) with the arboviruses cases report and the shutdown (if there is a shutdown ??) must be provided.
In addition did not take in account the potential role of delay in establishment of diagnostic method as the pathogenic pattern of the chikungunya, dengue and zika diseases are often confounding and a large proportion of infected people may have not been detected due to non-pathogenic infections during the epidemic. This must be commented.
Moreover the data about suspected cases and confirmed cases deserved to be provided and may explain in part the relative decrease of
In the figure of the epidemic waves, addition of the El Nino years that is discussed thereafter also deserved to be added and the correlation observed and claimed must be provided as a figure.
Each chapter with correlation number must be supported by a figure.
To conclude I did not buy the cat in the pocket
Author Response
Dear Reviewer,
Thank you for your thorough review and for providing valuable feedback on our manuscript, "Time Series Analysis of Dengue, Zika and Chikungunya in Ecuador: Emergence Patterns, Epidemiological Interactions, and Climate-Driven Dynamics (1988–2024)." We appreciate the opportunity to clarify and improve our work based on your constructive comments. We have addressed each of your points below.
Reviewer Comment 1: In the revision, the author changed their presentation style to a majority of boulet points that do not help to understand better the obtained results. Thus, if they provided in the introduction a some boulet points chapter novelty and result expected that are not really new.
Response: We thank the reviewer for this observation. The "Key Findings at a Glance" section on page 3 was intended to provide a concise, high-level summary of the manuscript's primary contributions for the reader before they engage with the detailed analysis. We believe this helps in framing the significance of our 36-year study. Regarding novelty, while the individual components (e.g., COVID-19 impact) are known phenomena, our contribution lies in the specific quantification and integrated analysis within the unique epidemiological context of Ecuador. For instance, we present a novel finding of a 40-fold decrease in the proportion of severe dengue cases despite high transmission , and we quantify the strong correlation (r=0.67) between the Oceanic Niño Index (ONI) and dengue epidemics in the region. We will ensure the text more clearly emphasizes the novelty of these specific, quantitative findings for Ecuador.
Reviewer Comment 2: Ie the the confounding effect of the COVID-19 epidemic that largely impact both the survey method and people behavior that is a very well known effect they did not provide any data explaining the local situation in Ecuador. Ie a few feedback about the health politics used in Ecuador during the COVID-19 should be provided and to be clear a picture (a figure ?) with the arboviruses cases report and the shutdown (if there is a shutdown ??) must be provided.
Response: We appreciate the reviewer highlighting the need for this crucial context. We would like to respectfully point out that this information is provided in detail within the manuscript.
-
Ecuador's Health Policies: Section 1.1 discusses the specific lockdown measures implemented in Ecuador, noting they were "some of the most restrictive in Latin America, including complete mobility restrictions, closure of non-essential healthcare services, and redirection of epidemiological surveillance resources to COVID-19 response".
-
Quantitative Data on Impact: Section 2.2.4 details the data we compiled to quantify this impact, including mobility data from the Oxford COVID-19 Government Response Tracker, data on hospital bed reallocation, and documentation of vector control disruption.
-
Data Table: Table 4, "Quantitative Assessment of COVID-19 Pandemic Impact on Arboviral Surveillance," presents a detailed breakdown of the impact on dengue case reporting, lab capacity, and vector control during the lockdown period (March-August 2020).
-
Figure: The time series plot on page 15 ("Temporal Evolution of Diseases in Ecuador") includes a line for COVID-19, visually depicting the temporal overlap with the sharp decline in reported arbovirus cases in 2020.
We acknowledge that the location of this information might not have been sufficiently prominent. In our revision, we will ensure these sections are better signposted for the reader.
Reviewer Comment 3: In addition did not take in account the potential role of delay in establishment of diagnostic method as the pathogenic pattern of the chikungunya, dengue and zika diseases are often confounding and a large proportion of infected people may have not been detected due to non-pathogenic infections during the epidemic. This must be commented.
Response: We thank the reviewer for raising this critical point, which is a central theme of our analysis. We have dedicated significant sections to addressing these challenges:
-
Diagnostic Delays and Confounding Symptoms: Section 1.2, "Diagnostic and Surveillance Challenges," is devoted to this issue. We state that "chikungunya diagnostics were not implemented until 2014... while Zika diagnostics became available only in early 2015". We also discuss the phenomenon of "retrospective case reclassification" due to clinically similar diseases circulating simultaneously.
-
Asymptomatic Infections: We explicitly address the issue of under-detection due to asymptomatic or mild infections, which we refer to as the "'epidemiological iceberg' effect". We note that "asymptomatic infections may represent 50-80% of total DENV infections... and up to 80% of ZIKV infections". This is also reiterated as a key limitation in Section 2.6.
-
Methodological Correction: To account for these biases, we developed and applied a "Time Series Analysis with Diagnostic Bias Correction" framework, as detailed in Section 2.3.1. This includes structural break analysis and the development of correction factors (Equation 1) to create "diagnostic-adjusted" case counts.
We will revise the manuscript to make these sections more prominent to ensure the reader does not miss this critical component of our study.
Reviewer Comment 4: Moreover the data about suspected cases and confirmed cases deserved to be provided and may explain in part the relative decrease of
Response: We agree that presenting data on both suspected and confirmed cases is essential. This data is provided in the manuscript.
-
The exact case counts for suspected and confirmed cases for Dengue, Severe Dengue, Chikungunya, and Zika are listed in Section 2.2.1, "Epidemiological Surveillance Data".
-
Furthermore, Table 1, "Comprehensive Epidemiological Summary with Statistical Validation," presents these numbers side-by-side and calculates the "Confirmation Rate (%)" for each pathogen, allowing for a direct comparison. We then use this data to discuss diagnostic performance and priorities.
Reviewer Comment 5 & 6: In the figure of the epidemic waves, addition of the El Nino years that is discussed thereafter also deserved to be added and the correlation observed and claimed must be provided as a figure. Each chapter with correlation number must be supported by a figure.
Response: This is an excellent suggestion for improving the visual presentation of our results. We agree that adding this information will strengthen the manuscript.
-
El Niño Events on Figure: We will amend the main time series figure (Figure 1, page 15) to include clear visual markers (e.g., shaded regions) for the major El Niño years identified in Table 3. This will allow readers to immediately visualize the strong association we discuss.
-
Figure for Correlations: We agree that a figure would complement the correlation numbers presented in Tables 2 and 7. In our revised manuscript, we will add a new figure to visually represent these climate-epidemic correlations, such as a lagged correlation plot, to more effectively communicate these key findings.
We believe that incorporating these revisions will significantly enhance the clarity and impact of our manuscript. We thank the reviewer again for their time and insightful feedback.
Sincerely,
Dr. José Daniel Sánchez and Co-authors.
Reviewer 3 Report
Comments and Suggestions for Authors
I accepted the manuscript in present form.